# Verification of Angular Response of Sky Quality Meter with Quasi-Punctual Light Sources

**DOI:** 10.3390/s21227544

**Published:** 2021-11-13

**Authors:** Mirco Bartolomei, Lorenzo Olivieri, Carlo Bettanini, Stefano Cavazzani, Pietro Fiorentin

**Affiliations:** 1CISAS—Center for Studies and Activities for Space “Giuseppe Colombo”, University of Padova, Via Venezia 15, 35131 Padova, Italy; lorenzo.olivieri@unipd.it; 2Department of Industrial Engineering, University of Padova, Via Venezia 1, 35131 Padova, Italy; carlo.bettanini@unipd.it (C.B.); pietro.fiorentin@unipd.it (P.F.); 3Department of Physics and Astronomy, University of Padova, Vicolo dell’Osservatorio 3, 35122 Padova, Italy; stefano.cavazzani@unipd.it; 4INAF-Osservatorio Astronomico di Padova, Vicolo dell’Osservatorio 5, 35122 Padova, Italy

**Keywords:** sky quality meter, light pollution, sky brightness

## Abstract

Sky Quality Meter (SQM) is a commercial instrument based on photometers widely used by amateur astronomers for skyglow measurement from the ground. In the framework of the MINLU project, two SQM-LE units were integrated in an autonomous sensor suite realized and tested at University of Padova for monitoring light pollution from drones or sounding balloons. During the ground tests campaign before airborne measurement, the performance of both SQM units was verified in laboratory using controlled light sources as a reference input; the results showed that both units presented an angular response deviating consistently from the expected performance and that the sensors’ field of view was larger than the one declared in the manufacturer’s datasheet. This aspect in particular would affect direct skyglow measurements during flight as light sources close to the boundaries of the field of view would not be attenuated but instead detected by the sensors. As a direct consequence, the measurement of low-intensity skyglows at stratospheric altitudes could be affected by high-intensity punctual sources acting as lateral disturbances. A dedicated test campaign was therefore conceived and realized to investigate SQM unit response to light sources in the field of view and identify the true angular response curve; the setup consisted in a controlled rotatory stage moving the unit in front of a fixed diffusive light source. Different test conditions were used to validate the experimental procedure, demonstrating the repeatability of the measurements. This paper presents the experimental campaign and the resulting SQM angular response curve; results indicate for both SQMs a larger than expected field of view and the presence of a double peak in the angular response, which is likely related to a non-perfect alignment of SQMs collimation optics. Furthermore, the wider resulting curves suggest that the contribution of lateral sources is more prominent with respect to the response predicted by the manufacturer. For this reason, the utilization of baffles to restrict SQMs field of view is analyzed to minimize the disturbance of lateral light sources and two different geometries are presented.

## 1. Introduction

Light pollution has always been identified as an issue for astronomical observation [1]; in addition, recent studies underline the negative effects both on wildlife [2] and human health [3]. In this context, dedicated instruments have been employed to measure skyglow [4]; among them, Sky Quality Meters are extensively used around the world, typically by amateur astronomers, to quantify the skyglow aspect of light pollution, and measured data are usually shared on the net building an important database of observation data [5,6,7].

In this context, the MINLU (Misuratore di INquinamento Luminoso—light pollution measurement system) autonomous sensor suite [8] was developed by the University of Padova for monitoring light pollution from drones or stratospheric sounding balloons. Among the instrumentation on board of MINLU, two Sky Quality Meters (SQMs) are employed to directly measure skyglow. A stratospheric flight of the MINLU payload was successfully conducted on 8 July 2021, using the flight platform UniPiHAB04 by Space Systems Lab from University of Pisa, to collect data on light pollution in central Italy [9].

The flight platform was composed by a commercial latex 1200 g balloon filled with industrial-grade helium and a dedicated flight train including a 1.5 m diameter parachute implementing an architecture similar to the one used for other successful High-Altitude Balloon flights by University of Pisa [10]. 

MINLU payload was subjected to extensive ground testing before being employed for field measurement. During such tests it was observed a consistent deviation between the SQMs angular response and the expected one. In general, SQMs are calibrated by the manufacturer with a uniform diffuse light source, to relate the instrument output to a precise value of magnitude. In addition, the manufacturer declares an angular response curve with a single peak centered at 0° and a Full Width at Half Maximum (FWHM) of about 20° (see Figure 1), based on the test performed on a SQM-L unit by Cinzano in 2007 [11,12].

In the contest of MINLU ground testing, it was observed that the sensors’ field of view was larger than the theoretical one reported in the datasheet; this deviation was represented by a larger experimental FWHM value [13]. Skyglow measurements can be strongly affected by this behavior, as the sensor would detect light sources close to the boundaries of its field of view instead of cutting them down. In particular, in case of low-intensity skyglow measurements, high-intensity punctual sources such as street lights or astronomical objects acting as lateral disturbances would strongly affect the SQM reading. For this reason, a dedicated test setup was developed to evaluate the SQMs true angular response curve.

The paper will describe the implemented setup and present the test results: the study will underline how the use of a corrected angular response curve may strongly improve measurements for observing all-sky brightness evolution. Section 2 will introduce the test setup, while Section 3 will focus on the collected data and its discussion. Section 4 will introduce the utilization of baffles to mitigate the influence of lateral punctual light sources.

## 2. Test Description

The test for SQM angular response was performed with a simple but effective setup. The SQM was mounted on a rotation stage; the rotation axis was aligned with the instrument optical window and the light source was placed in front of the unit under test. Different sources were employed: a white LED, a collimated light and a Liquid Crystal Display (LCD), all with a circular window, were placed alternatively in front of the unit under test. To avoid disturbances, stray lights, and diffusion effects from the facility surfaces, a fixed mask was designed and inserted in front of the SQM, aligned with the light source. A sketch of the test setup is reported in Figure 2 while Figure 3 presents pictures of the SQM mounted on a rotatory stage with and without the mask.

Tests were executed scanning in several directions on a plane orthogonal to the rotation axis of the SQM; the setup allowed four operative configurations for the SQM, allowing flipping the instrument on its four lateral faces while maintaining the rotation stage axis aligned with the optical window. For sake of simplicity, the four operative configurations were defined as UP, DOWN, LEFT, and RIGHT (see Figure 4).

To measure the rotation commanded to the rotary stage, a laser was mounted on the top of the SQM unit, while a graduated ruler, with a resolution of 1 mm, previously levelled with the laser rotation plane, was placed 58.5 cm behind the setup (the distance was constrained by the facility room dimensions). This setup allowed an accurate determination of the rotation stage angle: considering a maximum uncertainty of 0.5 mm in laser spot position determination (derived by ruler resolution), the expected uncertainty in angle determination is less than 0.1 deg when the SQM is aligned to the light source (0 deg), decreasing with the rotation reaching less than 0.02 deg at ±60 deg (see Figure 5). The uncertainty related to the rotation angle could therefore be assumed to be at maximum 0.1 deg for the rotary stage angle determination.

A total of 12 tests were performed in the experimental campaign, varying the SQM model, the light source and its size, and the SQM orientation, as summarized in Table 1. For each test configuration a full angular scan was executed starting from the perfect alignment (θ = 0 deg) through incremental steps up to maximum angle (θ = 60 deg), and back to the initial alignment, repeating measurements at the same angular steps. The scan was then repeated in the other angular direction (θ = 0 to −60 deg). For each sampling step static measurements were performed by waiting output stabilization (controlling at least 10 equal consecutive readings).

Once the acquisition was completed the test setup was updated to the next investigated configuration.

The first three tests aimed to define the influence of the illumination conditions on the SQM response; as light source an LCD and an LED, both with a diffuser, consisting in a piece of white Nylon 2 mm thick, were used. The diffuser was employed to produce a uniform emission with no preferential emission direction of the light source. Two different diameters (10 and 18 mm) were used for the circular window of the LED, to assess the influence of the window size on the SQM response. For the LCD only the 18 mm window was employed. Once the sensitivity to different illumination conditions was investigated, the SQM-LE (serial number 5009) was operated pointing to a LED light source with an intermediate window size (15 mm), using four different orientations (UP, DOWN, LEFT, and RIGHT). In particular, test 4 (UP configuration) focused on the repeatability of the measurement: three different acquisitions were performed and the measurement’s variation evaluated. Tests 8 and 9 focused on the response of the other SQM-LE (serial number 5143); only two acquisitions instead of four were performed (UP and LEFT direction), as tests on previous SQM-LE confirmed that the elaborated curves for the respective DOWN and RIGHT direction can be obtained by mirroring the UP and LEFT curves Test 10 was performed with a collimated light source to assess any difference in the instrument response with respect to non-collimated lights. Last, tests 11 and 12 studied the utilization of baffles with different geometries to narrow the angular response of the SQM.

All results presented have been elaborated normalizing the unit output with respect to the magnitude resulting at 0 deg of misalignment employing the following formulation:(1)MNORM=2.512^(SQM0 deg−SQMα)
where $SQM_{0\;deg}$ is the instrument reading at 0 deg and $SQM_{\alpha }$ is the reading at the considered angular value.

## 3. Test Results

In the following subsections the main results from the tests conducted on both SQM units are summarized.

### 3.1. Test with Different Illumination Conditions

As reported in [10], data in Figure 6 show that the different light sources used in the three tests, as well as the windows size, do not influence the shape of the experimental curves (i.e., the response of the SQM under analyses); this confirms that the size of the windows chosen for the test is adequate to consider them as punctual light sources.

Focusing on the shapes of the calculated curves, two different peaks can be observed and will be discussed in the next subsection. In addition, just as a reference, a FWHM value close to 30° was obtained from the SQM angular response curve.

An additional test was performed with a collimated light source (test 10) to confirm that the employed LED source can be considered punctual. Figure 7 shows the comparison between the LED source (test 8, configuration UP) and a collimated light (test 10, configuration UP) for SQM 5143. It can be noted that the two curves are identical, confirming that as long as the source can be considered punctual, the angular response measurements are repeatable.

These preliminary experimental results suggest that the shape of the angular response curve is not fully consistent with the one declared in the manufacturer’s datasheet. In particular, the curve presents two peaks with slightly different magnitude and a larger than expected Full Width at Half Maximum (FWHM) value: ≈30° instead of the expected ≈20° [11]. Next subsections will focus on the investigation of the angular response curves for SQM-LE 5009. In order to verify if the issue was common to several SQM-LE units or limited to SQM-LE 5009 (tests 4 to 7), tests were also performed for the SQM 5143 (tests 8 and 9).

### 3.2. Test with Different Unit’s Orientation (SQM-LE 5009)

The characterization of angular response SQM-LE 5009 and measurement repeatability was addressed by conducting three different acquisitions between −60 deg and + 60 deg with the instrument in all four considered orientations; results can be seen in Figure 8 and Figure 9. For each angular step the measurement mean value and associated uncertainty were evaluated; the overall measurement error was calculated by propagating this uncertainty with the Kline-McClintock formula applied to Equation (1) [14]. Figure 8 (bottom) reports the relative uncertainty (i.e., divided for the normalized magnitude) for each measurement; it can be noted that the maximal error is always less than 4%. Results suggest that the SQM can provide repeatable readings and can be used to make punctual measurements. In addition, the resulting curve is consistent with the one reported in Figure 6, with two peaks and a larger than expected FWHM value.

Starting from the two orthogonal angular response curves obtained by the tests 4, 5, 6, and 7, a 3D-mapping of the angular response of the SQM 5009 was produced, as reported in Figure 10 (left). This volume was obtained as a solid of revolution having as a section the two angular response curves reported in Figure 8 and Figure 9, corresponding to two orthogonal planes aligned with the optical axis. The evaluation of the average FWHM, that can be seen in Figure 10 (right), was performed by sectioning the solid at the half of the peak value, and calculating it as the mean value of 10 different diameters; the resulting value is 26.4 ± 1.2 deg.

### 3.3. Test with Different Unit’s Orientation—SQM-LE 5143

Similar tests were performed on the SQM-LE 5143, to evaluate if the behavior determined for SQM-LE 5009 was repeatable or was associated with a random defect of that instrument. Figure 11 reports the angular response of SQM-LE 5143 for orientations UP (left) and LEFT (right); the other orientations were not tested, as the angular response was demonstrated to be symmetric. It can be noted that the double peak and the larger FWHM are replicated; in this case both curves show peaks with different magnitudes.

Figure 12 shows the results of 3D mapping of the SQM 5143 angular response obtained using the procedure used for SQM 5009. In this case, the calculated average FWHM is 31.1 ± 0.4 deg.

The test of unit 5143 confirms that the misalignment of components causing deviated angular response is common to both available SQMs.

### 3.4. Evaluation of Measured Sky Brightness in Case of Not Nominal Angular Response

In order to compare the collected results from the tests with the curve declared by the manufacturer, three solids of revolution were modelled, representing the 3D mapping of the angular response of the two tested SQM and of an instrument with a reconstructed response resembling the performances declared on manufacturer datasheet reported in Figure 1. Figure 13 shows the three solids, from the left for SQM 5009 (blue), the manufacturer equivalent one (red) and SQM 5143. As previously mentioned, all SQMs are calibrated by the manufacturer by placing them in front of a known uniform light source; a specific calibration offset is then assigned at each instrument, to uniform all measurements to the expected brightness value. Geometrically speaking, this is equivalent to scale up or down the height of the solids of revolutions, in order to obtain for all of them the same volume, that is proportional to the total amount of light collected by the instruments.

This geometrical representation allows comparing the response of different instruments at the same values of field of view (FOV). In more details, it is possible to evaluate the three solids volume at different FOVs and deduct the SQMs general response; this is performed by verifying which different values of FOV contribute more to light collection and therefore to the instruments’ final reading.

Table 2 shows the cumulative values of the collected light fraction for the three considered devices at different values of the FOV.

It can be seen that the behavior of the two tested SQMs deviate considerably from the declared one: the light collection is lower for smaller FOVs (up to 30 deg) and higher for larger FOVS (from ±20 deg up to ±60 deg).

This result suggests the following considerations: for the smaller values of FOV, SQMs 5009 and 5143 are less sensitive to the incoming light than expected: up to ±10 deg, the reduction in light collection is respectively 12.1% and 26.3%. It shall be noted that this reduction in the instrument sensitivity could consistently affect the capability of the SQM to measure the real brightness of the area of interest placed in front of it.

For all SQMs (including the one modelled on the manufacturer datasheet) only the 40% of the collected light comes from the central part of FOV (±10 deg); the remaining 60% is collected from lateral directions.

Considering that the area of interest for which the brightness measurements are executed is usually placed in front of the instrument, it is clear that a SQM is not the most suitable instrument for this kind of application. To overcome these limitations and reduce the higher sensitivity to lateral disturbances typical of instruments with a FWHM wider than declared (e.g., due to a non-perfect alignment of internal optics), it is necessary to place a baffle in front of the SQMs optics to reduce its field of view. In the next section the results of two measurements with two different baffles, with FOV respectively limited to 15 deg and 30 deg, will be discussed.

## 4. Influence of Baffles on Angular Response

The effect of using baffles with the SQM units has been investigated in tests 11 and 12 by designing and integrating two different baffles on SQM 5143, limiting its field of view respectively to 15 deg and 30 deg. Figure 14 reports the response of SQM 5143 without (blue line) and with the two different baffles (see Figure 15), the one with a field of view of 15 deg (left, black line) and the one of 30 deg (right, red line).

In both cases it can be noted that the baffles can limit the influence of light sources outside their field of view; due to the non-perfect symmetry of the curve it can be noted that the response in also in this case not completely symmetric. 

Adding a baffle generates a constant offset in instrument output by cutting the contribution of the incoming light from angles greater than the designed baffle’s field of view. To correct the output from the instrument, the value of the offset corresponding to the employed baffle shall be added to manufacturer’s calibration data and set into unit internal electronics which implements a field-upgradeable firmware. The offset values can be calculated using the 3D modelization of the device’s angular sensitivity: offsets represent the ratio in magnitudes between the volume inside the FOV of the employed baffle and the total volume of the solid.

Table 3 presents the calculated offset values to be subtracted to instrument reading in function of the baffle FOV, for the two tested units and the nominal one reconstructed from the manufacturer datasheet.

Alternatively, the same result may be obtained by executing two measurements (with and without the baffle) under a uniform light, obtaining the experimental offset that can be applied to the device readings.

## 5. Conclusions

In this paper the angular response curves of two off-the-shelf Sky Quality Meter LE units have been calculated through a dedicated test campaign. The elaboration of test data confirmed that the units operated with good repeatability under different light sources, but evidenced that the normalized angular response curve is different from expected showing a double peak instead of a single one and a larger than expected FWHM value. The sensitivity to different unit’s orientations was also investigated and results allowed to correlate the angular response deviation to a non-perfect alignment of the SQM internal optics.

To evaluate the effect of the elaborated angular response on instrument output, 3D maps of the SQM angular sensitivity were generated and compared with one modelled on the expected behavior from manufacturer datasheet.

The comparison of experimental angular responses suggest that the contribution of light sources close to the boundaries of the instrument field of view, theoretically negligible, could in fact strongly increase uncertainty in the scientific data. This could in particular affect measurements of sky brightness from stratospheric balloons, where attitude is not controlled and high-intensity punctual sources such as street lights or astronomical objects can act as lateral disturbances. To mitigate this behavior, the SQM unit’s performance was investigated mounting two dedicated baffles (with FOV respectively of 15 deg and 30 deg) in front of the input optics; the obtained angular response curves with two different baffle geometries show that field of view can be restricted to minimize interference of undesired light sources but, as expected, unit response remains asymmetrical. In addition, a numerical method based on the 3D sensitivity maps is suggested to evaluate the offset introduced by baffles and to correct the output data from the SQM units to account for the limited field of view used in the measurement.

## Figures and Tables

**Figure 1 sensors-21-07544-f001:**
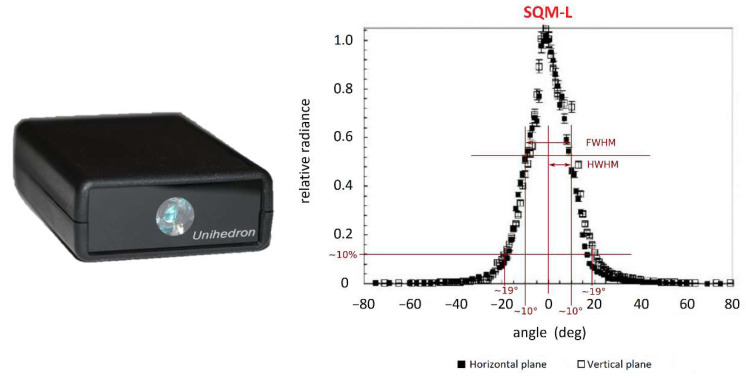
Image of SQM-LE unit (**left**) and manufacturer’s declared calibration curve (**right**) [11,12].

**Figure 2 sensors-21-07544-f002:**
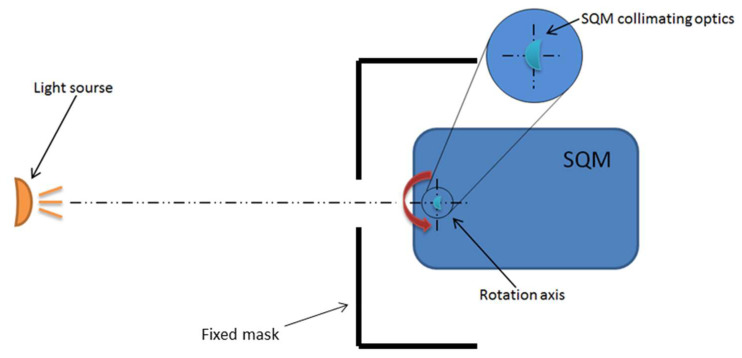
Sketch of test setup, highlighting the position of the rotation axis, centered on the SQM collimating optics.

**Figure 3 sensors-21-07544-f003:**
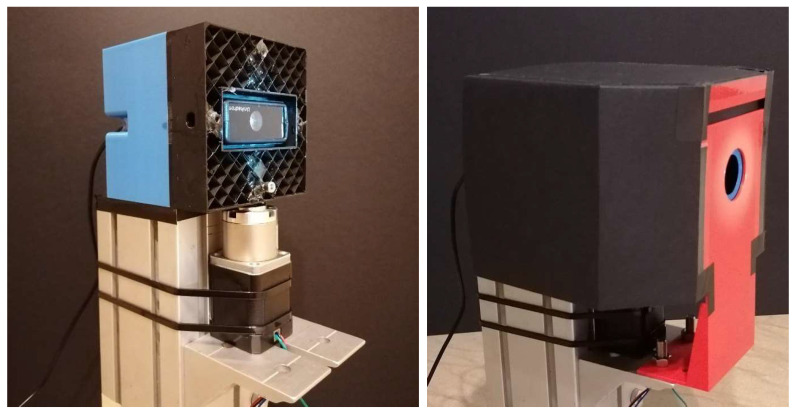
Test setup with the SQM mounted on a rotatory stage (**left**) and with the fixed mask shielding stray lights and diffusion sources (**right**).

**Figure 4 sensors-21-07544-f004:**
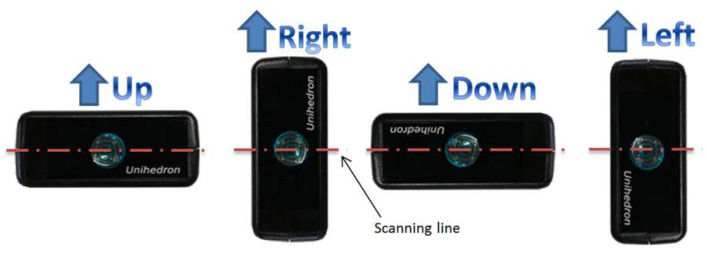
Overview of SQM unit orientation during measurements.

**Figure 5 sensors-21-07544-f005:**
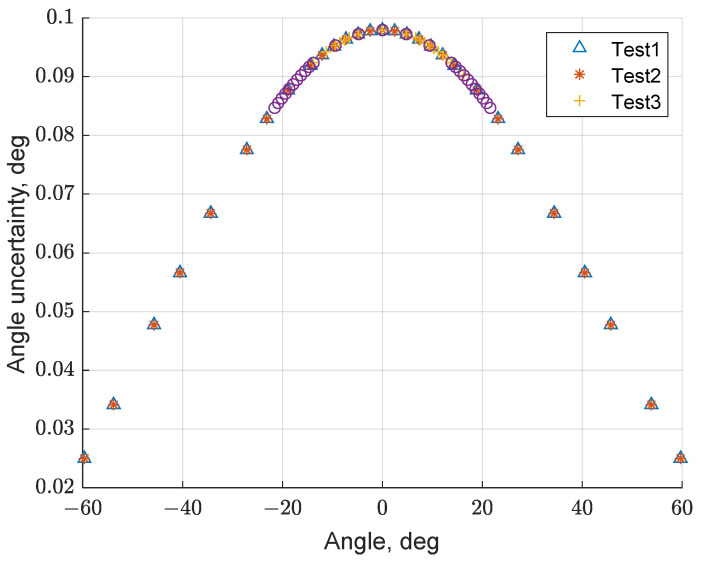
Uncertainty associated with angle measurement in function of the SQM rotation stage angle.

**Figure 6 sensors-21-07544-f006:**
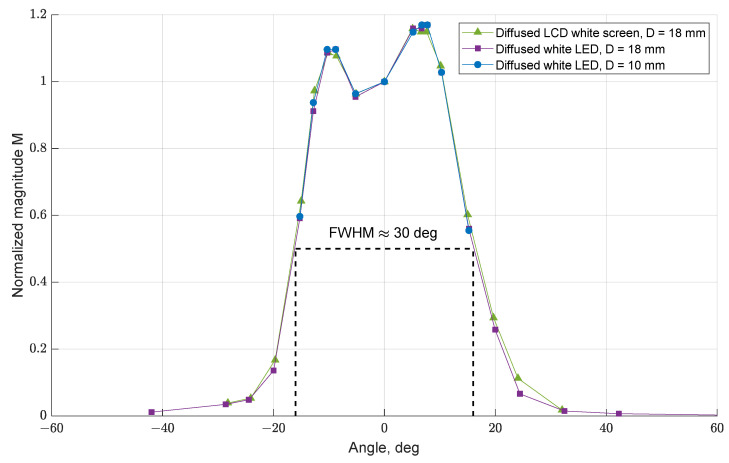
Normalized angular response curve for different illumination conditions (tests 1, 2, and 3, SQM-LE 5009).

**Figure 7 sensors-21-07544-f007:**
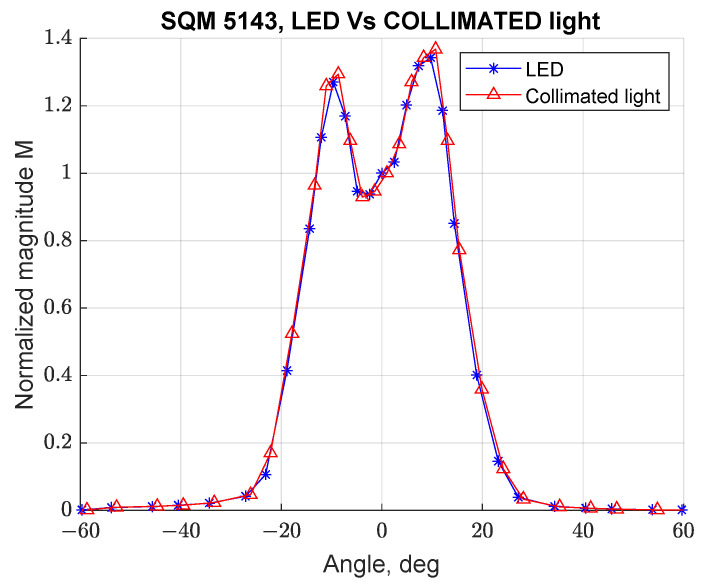
Comparison of normalized angular response curve for SQM-LE 5143 at different illumination conditions: LED in blue, collimated light in red.

**Figure 8 sensors-21-07544-f008:**
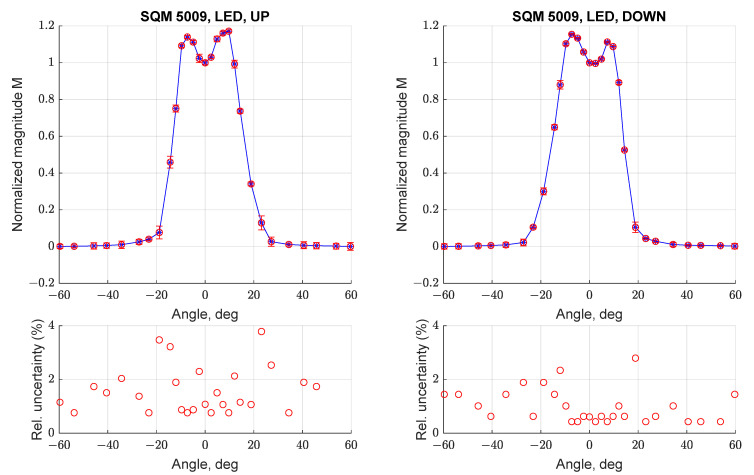
Normalized angular response curve (**top**) and relative uncertainty (1σ, **bottom**) for three scans performed with SQM-LE 5009 (UP and DOWN configuration).

**Figure 9 sensors-21-07544-f009:**
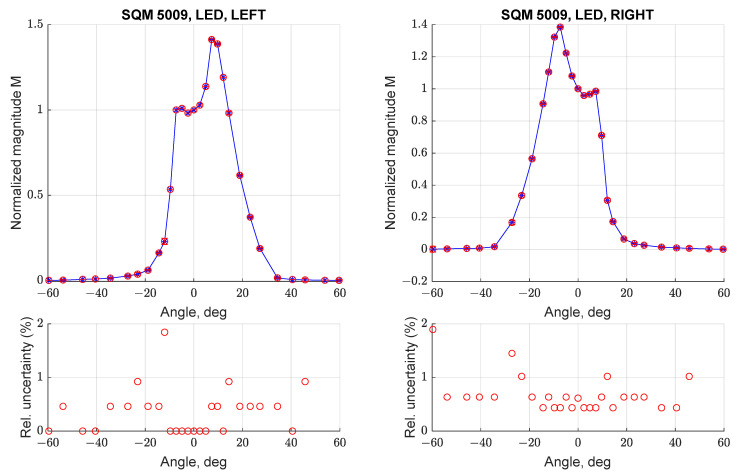
Normalized angular response curve (**top**) and relative uncertainty (1σ, **bottom**) for three scans performed with SQM-LE 5009 (RIGHT and LEFT).

**Figure 10 sensors-21-07544-f010:**
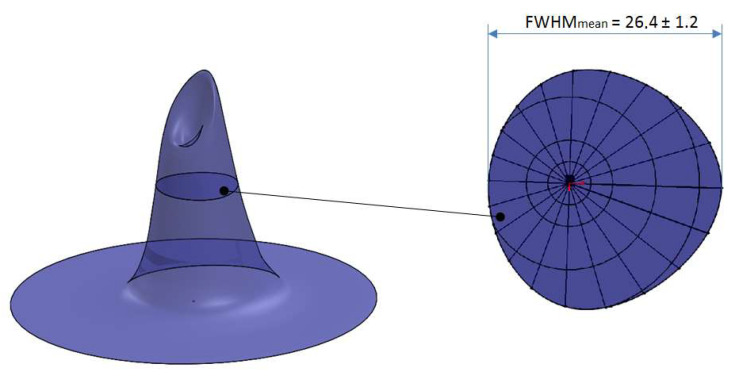
3-D mapping of SQM 5009 angular response (**left**), based on its two orthogonal angular response curves and average FWHM (**right**).

**Figure 11 sensors-21-07544-f011:**
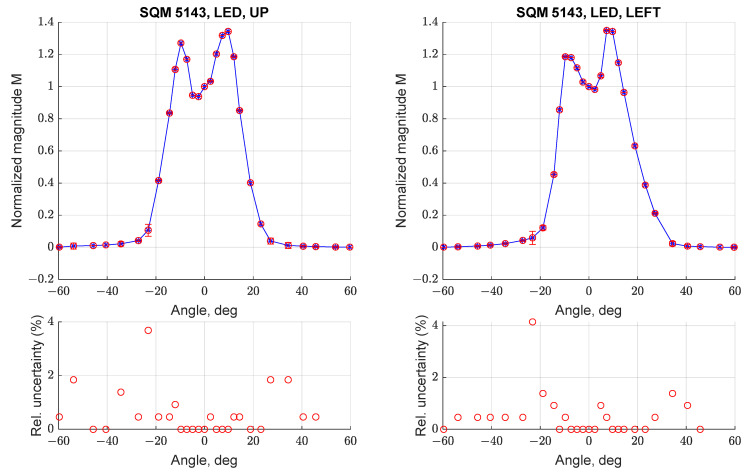
Normalized angular response curve (**top**) and relative uncertainty (1σ, **bottom**) for three scans performed with SQM-LE 5143 (UP and LEFT).

**Figure 12 sensors-21-07544-f012:**
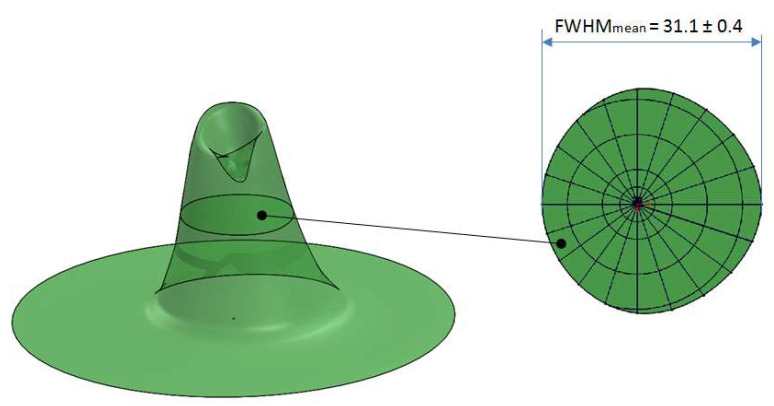
3-D mapping of SQM 5143 angular response (**left**) and average FWHM (**right**).

**Figure 13 sensors-21-07544-f013:**
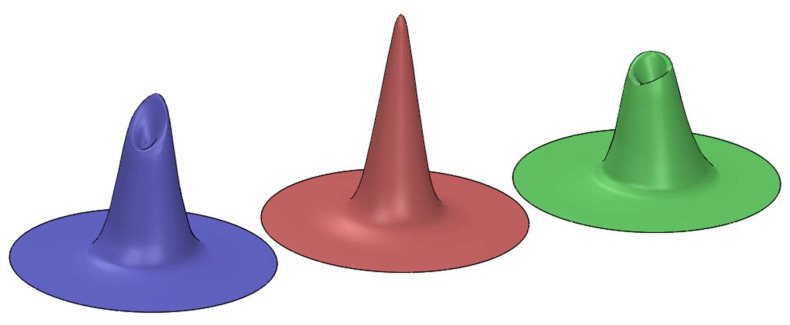
3-D mapping of angular response for SQM 5009 (**left**), reconstructed manufacturer (**centre**), and SQM 5143 (**right**).

**Figure 14 sensors-21-07544-f014:**
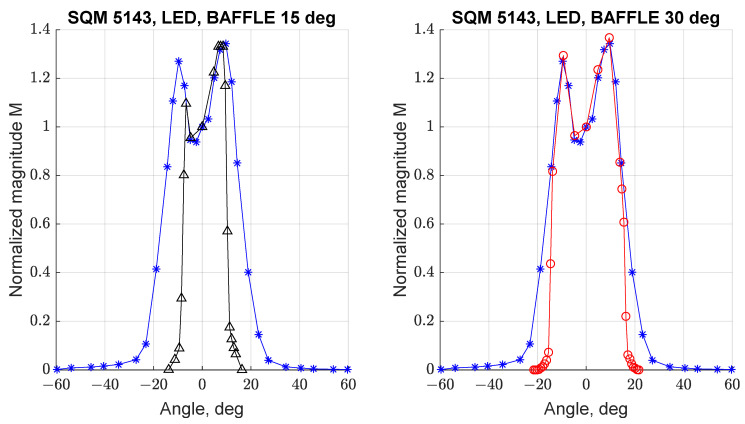
Effect of two different baffles on normalized angular response curves of SQM-LE 5143. On the left, 15 deg baffle (test 10, black line); on the right, 30 deg baffle (test 11, red line).

**Figure 15 sensors-21-07544-f015:**
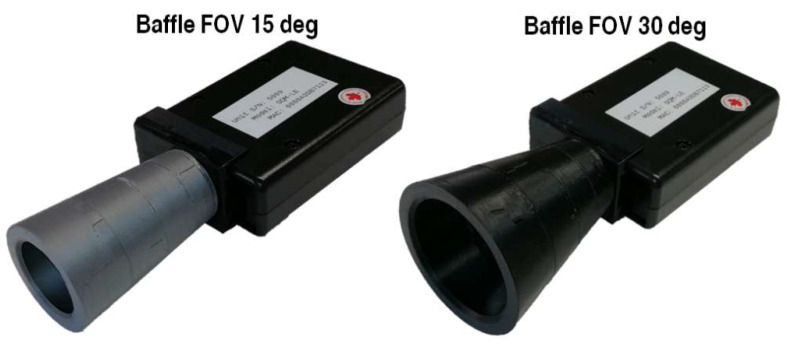
15 deg baffle (**left**) and 30 deg baffle (**right**) integrated on SQM unit.

**Table 1 sensors-21-07544-t001:** List of tests performed in the experimental campaign.

Test Number	SQM Model	Light Source	Window Size [mm]	SQM Orientation	Investigation:
1	5009	LCD	18	UP	Effect of different illumination conditions
2	5009	LED	18	UP	
3	5009	LED	10	UP	
4	5009	LED	15	UP	Effect of orientation—SQM-LE 5009
5	5009	LED	15	DOWN	
6	5009	LED	15	LEFT	
7	5009	LED	15	RIGHT	
8	5143	LED	15	UP	Effect of orientation—SQM 5143
9	5143	LED	15	LEFT	
10	5143	LED—Collimated	15	UP	Collimated light
11	5143	LED	15	UP	Baffle 15 deg
12	5143	LED	15	UP	Baffle 30 deg

**Table 2 sensors-21-07544-t002:** Cumulative collected light fractions for different FOV and variation from manufacturer model.

FOV [deg]	Collected Light Fraction [%]				
	EXPECTED (Reconstructed from Manufacturer Curve)	SQM 5009	SQM 5009 Variation from EXPECTED (%)	SQM 5143	SQM 5143 Variation from EXPECTED (%)
±2.5	3.4	2.0	−41.9%	1.5	−55.3%
±5	12.5	8.2	−34.6%	6.3	−49.5%
±10	39.0	34.3	−12.1%	28.7	−26.4%
±15	62.7	62.2	−0.9%	58.1	−7.4%
±20	75.8	80.3	5.9%	78.5	3.5%
±25	83.1	90.0	8.3%	89.0	7.1%
±30	88.0	94.6	7.4%	93.8	6.6%
±35	91.4	96.6	5.6%	96.0	5.0%
±40	93.9	97.7	4.1%	97.4	3.7%
±45	95.9	98.5	2.7%	98.3	2.5%
±60	99.9	99.9	0%	99.9	0%

**Table 3 sensors-21-07544-t003:** Offset values introduced by the baffles with different FOV on the expected angular response reconstructed from the manufacturer datasheet and for the two SQM units under analysis.

Baffle FOV [deg]	OUTPUT OFFSET on EXPECTED (Magn/arcsec^2^)	OUTPUT OFFSET on SQM 5009 (Magn/arcsec^2^)	OUTPUT OFFSET on SQM 5143 (Magn/arcsec^2^)
±2.5	−3.67	−4.26	−4.55
±5	−2.26	−2.72	−3.00
±10	−1.02	−1.16	−1.36
±15	−0.51	−0.52	−0.59
±20	−0.30	−0.24	−0.26
±25	−0.20	−0.11	−0.13
±30	−0.14	−0.06	−0.07
±35	−0.10	−0.04	−0.04
±40	−0.07	−0.03	−0.03
±45	−0.04	−0.02	−0.02
±60	0.00	0.00	0.00

## Data Availability

The data presented in this study are available on request from the corresponding author. The data are not publicly available as the official repository related to this project is still under development.

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
