# Peer review of "Verification of Angular Response of Sky Quality Meter with Quasi-Punctual Light Sources"

_sensors, 2021, doi:10.3390/s21227544_

Round 1

Reviewer 1 Report

The authors provide a measurement report on the directional sensitivity of an SQM device. It can be helpful, as SQM is used frequently for measuring night sky brightness. However, this device is not a perfect scientific instrument, as the manuscript also indicates. My primary question is whether this paper helps to obtain reproducible measurements by SQM devices. The manuscript demonstrates that there are problems with the device but provide no remedies.

First of all, the authors describe their procedure as a calibration (even in the title). However, this is not a calibration. Please refer to the "International vocabulary of metrology" (https://www.bipm.org/documents/20126/2071204/JCGM_200_2012.pdf/f0e1ad45-d337-bbeb-53a6-15fe649d0ff1)
According to this document:
"calibration:  operation that, under specified conditions, in a first step, establishes a relation between the quantity values with measurement uncertainties provided by measurement standards and corresponding 
indications with associated measurement uncertainties and, in a second step, uses this information to establish a relation for obtaining a measurement result from an indication."

Similarly, the curves presented in the paper give the angular response of SQM. "Calibration curve" has a very different meaning: "Calibration curve: 
expression of the relation between indication and  corresponding measured quantity value" 

The manuscript describes the deviation of the angular response from the published data. (Please note that the angular response displayed in Figure 1 is not the manufacturer's data but a figure from P. Cinzano publication - please refer to the original work.) However, these data alone do not provide information about the uncertainty of the measurements. It only gives an emergency signal, but not the solution. We know that the measurements cannot be compared with another measurement because of the angular sensitivity mismatch of the device. Thus really relevant information is missing from the paper.
Unanswered questions:
- If the device is recalibrated, what uncertainties result from the data? Have the authors made any recalibration of the device?
- How should the measurements be corrected?
- Is the misalignment of the optical element is stable - for example, can a mechanical shock during a balloon flight change the alignment of the internal optical components.
- A very important one: How the baffles affect the calibration (the relation of the read values to the real sky radiances
- What about other issues, like spectral mismatch, zero points correction and how they relate to the angular response mismatch?

Just a note about SQM: This device has lots of uncertainties. Is not there any alternatives for scientific grade measurements?

Some minor issues:

Line 47: July 8th - Please add the year, too.

Line 47 UniPiHAB04 - please describe it.

Reviewer 2 Report

The paper “Calibration of a sky quality meter with quasi-punctual light sources” by Mirco Bartolomei, Lorenzo Olivieri, Carlo Bettanini, Stefano Cavazzani and Pietro Fiorentin describes characterization of the angle responsivity of a commercial Sky Quality Meter - Lens Ethernet (SQM - LE) intended for the measurement of the darkness of the night sky from ground. Two SQM-LE units have been tested at University of Padova for monitoring light pollution from drones or sounding balloons in the frame of the light pollution measurement (MINLU) project. Before airborne measurements, the field of view (FOV) of both SQM-LE units has been tested in laboratory, and significant difference form the manufacturer’s specification revealed. The characterization set up consisted in a controlled rotatory stage moving the unit under test in front of a fixed diffusive punctual light source. Reasonably good repeatability was achieved by using five different light sources conforming so the reliability of results. For both tested instruments a larger than stated by manufacturer FOV and two peaks in the angular response were evident. The determined difference in stated and actual FOVs can strongly affect sky darkness measurements, as closely located high-intensity point sources can act as lateral disturbances.

Article is rather well written, but some terms used are not in conformance with the VIM/GUM terminology recommended in metrology. According to the International Vocabulary of Metrology (VIM), Calibration is an operation performed on a measuring instrument or a measuring system that, under specified conditions (1.) establishes a relation between the values with measurement uncertainties provided by measurement standards and corresponding indications with associated measurement uncertainties and (2.) uses this information to establish a relation for obtaining a measurement result from an indication. The objective of calibration is to provide traceability of measurement results obtained when using a calibrated measuring instrument or measuring system.

A key term for calibration is measurement uncertainty. A measurement result is an estimate of the value of the measurand and thus it is complete only when accompanied by a statement of the uncertainty of that estimate.

In this article, at least the uncertainty of determined Full Width at Half Maximum (FWHM) shall be presented. Only when compared with the stated for FWHM uncertainty one can see how significant is the difference between the manufacturer’s specification and the determined by authors FWHM.

I recommend that the article may be published after revision by authors providing relevant uncertainty estimate.

Round 2

Reviewer 1 Report

I accept the corrections in the paper and have no further comments/questions.

This manuscript is a resubmission of an earlier submission. The following is a list of the peer review reports and author responses from that submission.